# Multifunctional Core/Shell Diamond Nanoparticles Combining Unique Thermal and Light Properties for Future Biological Applications

**DOI:** 10.3390/nano13243124

**Published:** 2023-12-12

**Authors:** Sergey A. Grudinkin, Kirill V. Bogdanov, Vladimir A. Tolmachev, Mikhail A. Baranov, Ilya E. Kaliya, Valery G. Golubev, Alexander V. Baranov

**Affiliations:** 1International Research and Education Centre for Physics of Nanostructures, ITMO University, 197101 St. Petersburg, Russia; grudink@gvg.ioffe.ru (S.A.G.); mbaranov@itmo.ru (M.A.B.); kaliyailya2802@gmail.com (I.E.K.); a_v_baranov@itmo.ru (A.V.B.); 2Ioffe Institute, 194021 St. Petersburg, Russia; tva@mail.ioffe.ru (V.A.T.); golubev@gvg.ioffe.ru (V.G.G.)

**Keywords:** diamond, chemical vapor deposition, color centers, photoluminescence, temperature dependence

## Abstract

We report the development of multifunctional core/shell chemical vapor deposition diamond nanoparticles for the local photoinduced hyperthermia, thermometry, and fluorescent imaging. The diamond core heavily doped with boron is heated due to absorbed laser radiation and in turn heats the shell of a thin transparent diamond layer with embedded negatively charged SiV color centers emitting intense and narrowband zero-phonon lines with a temperature-dependent wavelength near 738 nm. The heating of the core/shell diamond nanoparticle is indicated by the temperature-induced spectral shift in the intensive zero-phonon line of the SiV color centers embedded in the diamond shell. The temperature of the core/shell diamond particles can be precisely manipulated by the power of the incident light. At laser power safe for biological systems, the photoinduced temperature of the core/shell diamond nanoparticles is high enough to be used for hyperthermia therapy and local nanothermometry, while the high zero-phonon line intensity of the SiV color centers allows for the fluorescent imaging of treated areas.

## 1. Introduction

Currently, local photothermal therapy is considered a promising method for the treatment of malignant neoplasms [1,2] and consists of the selective destruction of tumor cells via heating. In order to convert light energy into heat, nanoparticles with a high light absorption coefficient in the spectral region of tissue transparency are introduced into target tissues, and afterwards, under the influence of, for example, focused laser radiation, the tissues are locally heated to the required temperatures (40–46 °C). A main drive for the improvement of local hyperthermia techniques is the promise of effective, selective, and controlled heating. Due to the high absorption coefficient in the near-infrared region of the spectrum, diamond nanoparticles heavily doped with boron are very promising for photothermal therapy [3,4,5]. Additional advantages of the use of diamond nanoparticles as local photoinduced heaters are diamonds’ exceptionally high thermal conductivity, biocompatibility, and ease of bioconjugation thanks to well-developed surface functionalization techniques [6]. These properties of diamonds make them very attractive for biomedical applications. For instance, fluorescent nanodiamond–gold/silver nanoparticles were used for cell labeling and photothermal therapy [7,8]. In [9], laser-synthesized nanodiamonds coated with polydopamine exhibited strong photothermal properties combined with fluorescent imaging.

A variety of optical active color centers, whose emission wavelengths span a spectral range from the visible to the near-infrared, can be hosted in the diamond matrix [10]. This finding resulted in the creation of different luminescent diamond nanoparticles with embedded color centers highly suitable for all-optical local thermometry, bioimaging, and biosensing [11]. The former in this case is based on monitoring temperature-dependent changes in the photoluminescence intensity, spectral position, and linewidth of the zero-phonon line (ZPL) of the color centers [12,13,14,15,16,17,18]. It is important to note that nanoscale remote luminescence thermometry techniques can be applied to control the intracellular temperature reached during hyperthermia therapy. The most attractive color centers in diamond nanoparticles suitable for nanothermometry are the family of centers based on IV-group interstitial atoms (Si, Ge, Sn, and Pb) positioned between two vacancies in the diamond lattice [19]. Diamond nanocrystals with silicon vacancy (SiV) color centers, whose intense and narrow ZPL at ∼738 nm falls within the transparency region of biological tissues, are ideally suited for bioimaging, which is necessary for visualizing targets for hyperthermia therapy and local nanothermometry [5,12,15,16,17,18,19,20,21].

The idea to combine the advantages of the diamond nanocrystals doped by the different atoms in order to create a novel multifunctional heterostructure for efficient local photoinduced hyperthermia therapy, whilst also controlling the temperature of the tissue and labeling it with a fluorescent marker, looks very attractive [22]. At least two attempts to synthesize diamond nanocrystals doped with boron, silicon, and germanium are known [5,23]. In the first case [5], commercially available boron-doped nanodiamonds (BNDs) with an average size of 100 nm were implanted with silicon for optical temperature sensing. In the second case [23], BNDs were synthesized using the hot filament chemical vapor deposition (HFCVD) technique with GeV and SiV color centers embedded during the growth process.

In this paper, we report for the first time multifunctional core/shell diamond nanoparticles for local photoinduced hyperthermia therapy, thermometry, and fluorescent imaging manufactured using the HFCVD technique. Here, the core is a CVD diamond nanoparticle heavily doped with boron and heated by the absorbed laser radiation. The core, in turn, heats the shell, which is a transparent CVD diamond layer with embedded luminescent SiV color centers emitting an intense and narrowband ZPL and whose spectral position is temperature-dependent, allowing for precise thermometry. At acceptable laser radiation intensities, the temperature of nanoparticles induced by light can be high enough to be used for hyperthermia therapy and for local nanothermometry, while the high ZPL intensity allows for local fluorescent imaging.

## 2. Materials and Methods

### 2.1. The Fabrication of Luminescent Core/Shell Nanodiamonds (CSNDs)

The core/shell diamond nanoparticles were synthesized using the HFCVD method. A synthetic opal film on a fused silica wafer was used as a substrate on which the diamond nano- and sub-microparticles were fabricated. Synthetic opal films were formed from monodisperse spherical particles of amorphous silica (a-SiO_2_) of ~250 nm in diameter, closely packed into a face-centered cubic lattice using the vertical deposition method [24]. The thickness of the synthetic opal film was 9–15 monolayers. A technique proposed in Ref. [25] was used for growing CVD diamond nanoparticles (NDs) with a shape closest to spherical on the opal substrate. Figure 1a,b show typical scanning electron microscope (SEM) images of a single diamond nanoparticle and an ensemble of the nanoparticles, respectively, grown on the opal surface.

The low thermal conductivity of the opal film and the small contact area of the diamond particle with the substrate due to the close-to-spherical shape of the diamond nanoparticles make it possible to reduce heat leakage from the nanoparticle to the substrate and therefore decrease the cooling of the diamond particle during its heating by laser radiation. The nearly spherical shape of the diamond particles formed during CVD synthesis makes it possible to almost completely cover the surface of a diamond core heavily doped by boron with a diamond shell embedded with luminescent SiV color centers. It is important to note that the SiO_2_ substrate can be removed by etching with a solution of hydrofluoric acid (HF) and ammonium fluoride NH4F to obtain a colloidal suspension of diamond nanoparticles for a biomedical application [26].

In order to fabricate the multifunctional core/shell diamond nanoparticles, we developed a two-stage CVD synthesis process. At the first stage, the growth of diamond particles (diamond core) of about 800–1100 nm in size was carried out, during which doping with boron occurred by introducing diborane (B_2_H_6_) into the gas mixture. The boron-doped nanodiamonds (BNDs), used as the cores, were synthesized by the HFCVD method with parameters we previously determined [23]. Briefly, the substrate temperature was 800 °C, the temperature of the tungsten coil was 2000–2200 °C, the pressure in the reactor chamber was 48 Torr, the hydrogen flow rate was 480 sccm, the methane concentration was 4%, and the growth time was about 3 h. For the synthesis of highly absorbing BNDs, the ratio of boron atoms to carbon atoms (B/C) in the diborane–methane mixture was 64,000 ppm. The size of BNDs measured using scanning electron microscopy (SEM) ranged from 0.8 µm to 1.2 µm. An SEM image of a typical BND particle heavily doped with boron on the opal surface is shown in Figure 1c. Nanodiamonds of detonation synthesis with a size of ~4 nm applied by aerosol spraying onto the opal surface were used as nucleation centers [27], with a surface density of ~10^7^ cm^−2^.

At this stage, in order to control the effect of boron doping on absorption, a silicon plate was also placed in the CVD reactor chamber, on the surface of which, simultaneously with the formation of BNDs, a boron-doped diamond layer of about 1 μm in thickness was grown.

In the second stage, a transparent diamond shell containing luminescent SiV color centers was grown on the surface of the BND particles obtained in the first stage by using previously developed techniques [22]. The diamond shell was fabricated via the HFCVD method using the following process parameters: the temperature of the tungsten coil was 2000–2200 °C; the pressure in the chamber was 50 Torr; the hydrogen flow rate was 500 sccm; the methane concentration was 4%; the substrate temperature was 750 °C; growth time was 1.5 h. The source of the Si atoms was a crystalline silicon wafer situated on the substrate holder during the HFCVD growth process. During the HFCVD process, the etching of the solid-state sources of Si atoms with atomic hydrogen gives rise to the volatile radicals SiH_x_. The transfer and deposition of SiH_x_ on the surface of the growing NPs with the subsequent embedding of Si atoms into the diamond lattice led to the formation of SiV color centers.

An SEM image of a typical core/shell diamond particle with an absorbing core heavily doped with boron at 64,000 ppm in the gas mixture and a luminescent shell with formed SiV centers on the opal surface is shown in Figure 2a. As an illustration, a schematic image of a hybrid core/shell diamond nanoparticle is presented in Figure 2b.

### 2.2. Scanning Electron Microscopy, Photoluminescence, and Raman Setup

The scanning electron microscope (SEM) images of nanodiamonds particles with embedded boron atoms or/and SiV color centers as well as of the core/shell diamond particles were obtained with the Zeiss Scanning Electron Microscope, “Merlin” (Carl Zeiss, Ottobrunn, Germany), at an accelerating voltage of 10 kV at a probe current of 150 pA. To improve the image quality and topological contrast, the samples were fixed with carbon tape to create a conductive bridge between the silicon substrate and the sample holder, and the signals from the InLens and Everhart-Thornley SE2 detectors were recorded simultaneously.

The appearance of boron-induced absorption of the diamond particles was supported by comparison of the reflection spectra from the diamond layer grown in the same condition as the BNDs with an undoped diamond layer of the same thickness by using a standard setup of a 45-degree reflection.

The photoluminescence and Raman spectra of the nanodiamonds were measured using a Renishaw “InVia” Raman spectrometer (WITec, Ulm, Germany) equipped with a Leica confocal microscope, liquid-nitrogen-cooled CCD, and 1800 lines/mm grating providing ~2 cm^−1^ spectral resolution. The excitation laser radiation of 488 nm with power of 0.01 or 0.1 mW was focused by a 100× lens (NA = 0.9) into a spot with a diameter of ~1 µm on the single selected diamond particle. The corresponding laser radiation power density was about 0.3 kW/cm^2^ or 3 kW/cm^2^.

The correlation between the light-induced spectral shift of the SiV ZPL (local temperature of the core/shell NDs) and the incident light power was determined using a home-built setup. The setup allowed the incident laser beam to be focused on an ensemble of diamond particles in a spot with a diameter of ∼50 µm using a 50× lens with NA = 0.8 and to register the photoluminescence spectrum with a Princeton Instruments Action SP2500 triple grating monochromator (Princeton Instruments, Trenton, NJ, USA), equipped with a diffraction grating of 1200 grove/mm, allowing a 2 cm^−1^ spectral resolution. A CW argon-laser LCS-DTL (Laser-Export Co. LTD, Moscow, Russia) with a line of 532 nm was used for sample irradiation with variable power in the range of 0.1–55 mW, which corresponds to power density on the sample variable in the range of about 0.005–0.3 kW/cm^2^ in small steps.

To correctly measure the intensities of secondary emissions of boron- and silicon-doped diamond particles in a broad spectral range, they were normalized to the spectral sensitivity of the spectrometer determined preliminarily using a standard “black body” emission with “Stabilized Tungsten Light Source, SLS201(/M), (ThorLabs, Newton, NJ, USA)”. Since the used spectrometer allows both the luminescence and Raman spectra to be obtained simultaneously from the same individual diamond nanoparticle, for a comparison of the luminescence intensities obtained from nanocrystals of different sizes, the luminescence intensities were also normalized to the intensity of the diamond Raman line of ~1332 cm^−1^ (521.9 nm), which is proportional to the illuminated nanocrystal volume. All measurements were carried out at least 5 times with different diamond particles to prove the reproducibility of the data obtained.

## 3. Results

### 3.1. BNDs Characterization

#### 3.1.1. Raman Spectra

Figure 3 shows the Raman spectrum of a BND particle synthesized on the opal substrate and doped with boron at a content of 64,000 ppm in the CH_4_/B_2_H_6_ gas mixture in the vicinity of the fundamental Raman band of diamond at 1332.5 cm^−1^. It can be seen that the asymmetric spectral response typical of the Fano resonance dominates the spectrum of the BNDs. The appearance of such a feature in the Raman spectrum is a signature of doping diamonds with boron at concentrations in the gas mixture above several tens of thousands ppm, which leads to a change in the structure of the diamond crystal lattice [3,23,28,29,30]. The increase in the degree of boron doping leads to a shift in the Raman band of diamond from ~1332.5 cm^−1^ to low frequencies down to 1300 cm^−1^ and an increase in its FWHM from 6 cm^−1^ to ~20 cm^−1^. According to [29], the Fano resonance occurs due to the interaction of a discrete Raman transition with a continuum of electronic transitions due to the formation of metallic conductivity in the diamond lattice at a high doping level. The structural defects that appear in this case also lead to a violation of the wave vector selection rule in Raman scattering of diamond. This leads to the appearance in the broad bands of the Raman spectra (see Figure 3), with maxima in the region of 1230 cm^−1^ and 1205 cm^−1^, corresponding to peaks in the single-phonon density of states of diamonds. Similar changes observed in the Raman spectra of the BNDs synthesized by us indicate a high degree of doping of the diamond particles with boron during the HFCVD process.

#### 3.1.2. Absorption

The appearance of boron-induced absorption of the diamond particles at a high boron content was supported by comparison of the reflection spectra from a diamond layer grown in the same conditions as the BNDs with an undoped diamond layer of the same thickness. The reflection spectra were measured in s-polarization at an incident angle of 45 °C. In the reflection spectrum of the undoped diamond film (Figure 4a), interference fringes are present due to light reflection from two-plane parallel layer interfaces. Figure 4a illustrates the disappearance of the interferential spectral structure of the diamond layer grown and doped by boron at a content of 64,000 ppm in the gas mixture together with the studied BNDs. It follows that the introduction of boron atoms into the diamond lattice at a large boron content, as we implemented during the fabrication of BNDs, led to such strong light absorption by a 2 μm thick diamond layer that the corresponding interference fringes in reflection completely disappeared.

#### 3.1.3. Photoluminescence

Figure 4b demonstrate a secondary emission of a heavily doped BND particle grown on the opal substrate and excited by a 488 nm light which contains the Raman response at ~518 nm (~1215 cm^−1^, see, Figure 2) and a broad luminescence background. An approximately ten times weaker spectral intensity of the background luminescence as compared to that of Raman signal supports the quenching of the broadband luminescence as a result of the boron doping. The same phenomenon was observed earlier in Ref. [23].

### 3.2. Core/Shell NDs Characterization and Temperature Dependent Properties

#### 3.2.1. Raman and Photoluminescence of CSNDs with BND Core and Shell Doped by SiV Centers

Figure 5 demonstrates the Raman and PL spectra of a single core/shell diamond nanoparticle consisting of a heavily boron-doped diamond core of about 800 nm in diameter covered by a diamond shell with embedded luminescent SiV color centers of ~500 nm in thickness. The SEM image in Figure 2a shows a typical example of the studied core/shell diamond particle. Optical spectra were obtained by using a setup based on a Renishaw “InVia” Raman spectrometer.

The Raman spectrum presented in Figure 5a demonstrates several bands characteristic of the slightly disordered lattice of the diamond shell: an intense 1332 cm^−1^ band is characteristic of the diamond lattice with sp^3^ hybridization, the broadband at about 1500 cm^−1^ can be assigned to the G-band of amorphous carbon, while the shoulder at about 1560 cm^−1^ indicates the presence of the sp^2^ phase of carbons [31]. All of the bands belong to the diamond shell, while a weak broad Raman band of about 1220 cm^−1^ indicates the presence of the heavily boron-doped core in the studied core/shell ND. The PL spectra of the core/shell ND at two strongly different excitation power densities of 0.008 and 0.08 mW are shown in Figure 5b. The narrow ZPLs of the SiV color centers located in the shell dominate the spectra at wavelengths of 738.4 nm and 739.0 nm, respectively, demonstrating the power dependence of the ZPL spectral position of the SiV color centers. The inset in Figure 5b illustrates a ZPL shift of about 0.6 nm. It is noted that the dependence of the spectral position (energy) of the SiV ZPL on the excitation power is expected because of the light absorption and heating process in the BND core, since the excitation power controls the temperature, which in turn determines the SiV’s ZPL position [12,15,16,17,18,19,20,21]. A weak Raman signal of the shell can be recognized in the spectra in the region of about 518 nm.

A comparison between spectral positions of the ZPLs for core/shell ND (red line) and ND without boron doping (blue line) obtained at the same excitation power of 0.8 mW is presented in Figure 5c. It can be seen that the presence of a boron-doped core in a diamond particle leads to a red shift in the ZPL of the SiV centers located in the shell compared to undoped ND, which indicates an increase in the shell temperature when the core is heated due to the absorption of exciting radiation.

#### 3.2.2. T-Dependence of ZPL Spectral Position

The correlations between the incident-light-induced spectral shift in the SiV ZPL (in other words, the local temperature of the CSNDs) and the incident light power was determined using a home-built setup which focuses the 532 nm laser radiation to a spot with a diameter of ∼50 µm, allowing it to illuminate an ensemble of CSNDs. Moreover, the sample could be irradiated with slightly incremented light power in the range of 0.1–55 mW, covering the ND temperature range relevant for biomedical applications.

In Figure 6, we show a set of PL spectra in the region of the SiV ZPL at 738 nm excited by 532 nm radiation at different power levels. The red shift in the SiV ZPL peak with increasing incident light power is clear seen and was caused by the laser-induced heating of the CSNDs. The temperature-induced red shift is accompanied by the increasing ZPL intensity and ZPL broadening. The dependence of the spectral position of the SiV ZPL on incident light power in the broad range of 0.1–40 mW (5–2000 W/cm^2^) is shown in Figure 7a. A practically linear plot of the dependence up to a power around 5 mW (~250 W/cm^2^) can be seen, above which the dependence demonstrates evident saturation. There are well-established data on the temperature dependence of the SiV ZPL peak position [12,15,16,17,18,19,20], and using such data, it is possible to plot the dependence of the temperature of illuminated CSNDs on the irradiation power. In order to achieve this, we used the data reported in Ref. [17], where a temperature-induced ZPL shift was measured both in the temperature region used for photothermal therapy (25–55 °C) and for higher temperatures up to 600 °C [20]. A dependence of the ND temperature on the laser radiation power, calculated using the data from [20], is shown in Figure 6b. Here, as in the case of the dependence of the ZPL position on power, the linear dependence region occurs at a power of up to 5 mW (see inset in Figure 7b), and saturation takes place with the further power increase.

Analogous growth and saturation were observed dependent on ZPL intensity of the laser radiation power (Figure 7c). It should be noted that, usually, the ZPL intensity of SiV centers decreases with the increase in diamond temperature [12,17,18,19,20,21]; however, this is not the case here, since in the CSNDs under study, the ZPL intensity is determined by a competition between a decrease in intensity due to an increasing probability of non-radiative recombination and an increase in the signal intensity with the excitation intensity that results in a more intensive ZPL response. The reasons for saturation have not been established; most likely, the increase in heat dissipation into the substrate and the environment with the increasing temperature of the particle compensates for the increase in heating via laser radiation. As a result, a photoinduced temperature of the CSNDs above ~150 °C can be easily reached in the 50 µm spot at a 532 nm laser radiation of 40 mW.

Next, we shall compare our data with those obtained in [17] for the temperature range of 25–65 °C, which is important for photothermal therapy. The plots in Figure 7d of temperature versus ZPL position show that the experimental data obtained in the current work and in the work on dart-like diamond microneedles (DDMNs) by L. Golubewa et al. [17] are very close, despite the difference in the morphology of the diamond nanoparticles under study.

Thus, we found that the temperature of the CSNDs can be precisely manipulated by the power of the incident light, and the temperature relevant for thermotherapy of about 45–50 °C can be achieved with an incident light power of 1.1–1.3 mW or a power density of 44–52 W/cm^2^. The obtained values fall within the range of light power densities safe for biological systems reported in [32], where it was noted that “for practical purpose, optimum intensity depends on cell type, photothermal agents, bio-environment etc. and the efficient photothermal treatment varies in the laser range from 1 to about 100 W/cm^2^”.

## 4. Conclusions

The development of multifunctional core/shell CVD diamond nanoparticles for local photoinduced hyperthermia, thermometry, and fluorescent imaging is demonstrated. The heating of the core/shell ND up to 150 °C by the incident light of a 532 nm wavelength is observed due to presence a heavily boron-doped diamond core heated by the absorbed laser radiation. The heating was indicated by the temperature-induced spectral shift in the intensive ZPL of the SiV color centers embedded in the diamond shell. It is shown that at the incident light power safe for biological systems, the photoinduced temperature of the core/shell ND is high enough to be used for hyperthermia therapy and local nanothermometry, while the high ZPL intensity allows for fluorescent imaging of the treated areas. The results obtained can be considered as preliminary, demonstrating a proof of concept of the synthesis and application of the core/shell diamond nano- and sub-microparticles, with the core and shell possessing different functional properties for photoinduced hyperthermia, local thermometry, and fluorescent imaging due to being doped by different impurities. It is evident that this technology could be developed in more detail in order to increase its efficiency for biomedical applications. For example, the colloidal suspension of the CSNDs must be obtained by etching the SiO_2_ substrate with appropriate chemicals, and incident radiation at the wavelength that is not absorbed by biological tissues but absorbed by the core of the core/shell ND has to be used, e.g., light at a wavelength of 633 nm or heating via two-photon excitation.

## Figures and Tables

**Figure 1 nanomaterials-13-03124-f001:**
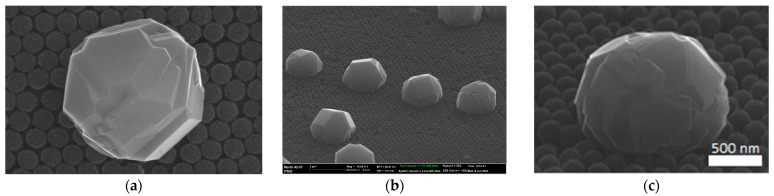
Typical SEM images of the HFCVD diamond nanoparticles grown on SiO_2_ opal surface: (**a**) a single diamond nanoparticle of ~1 µm in diameter; (**b**) an ensemble of the diamond nanoparticles with size of ~800 nm; (**c**) a single 1 µm diamond nanoparticle heavily doped with boron (BND particle).

**Figure 2 nanomaterials-13-03124-f002:**
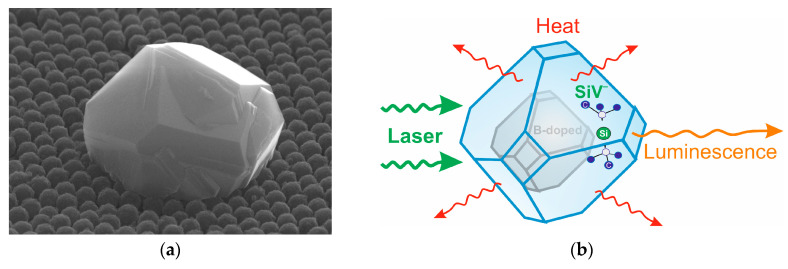
(**a**) SEM image of the core/shell diamond particles of ca. 1.3 µm in diameter with a core heavily doped by boron (64,000 ppm in the gas mixture) and a shell with luminescent SiV centers on an opal surface. (**b**) A schematic image of hybrid core/shell diamond particle with a boron-doped core absorbing laser light and a diamond shell with luminescent SiV centers.

**Figure 3 nanomaterials-13-03124-f003:**
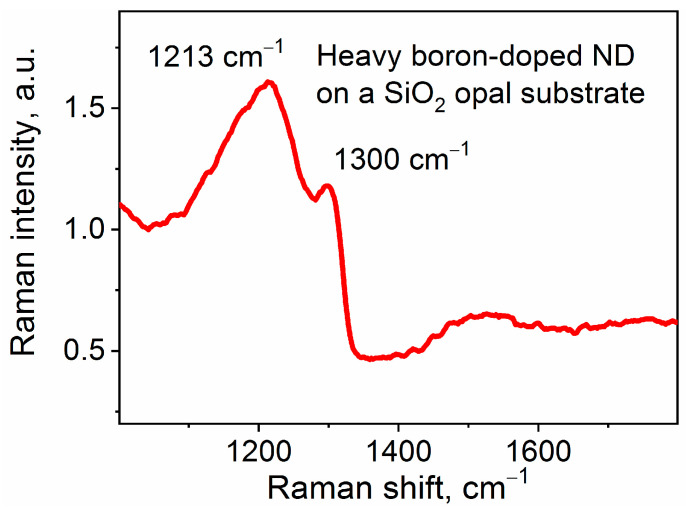
Raman spectra of a BND particle doped with boron at a content in the gas mixture of 64,000 ppm on the opal substrate. Bands of heavily doped diamond (~1300 cm^−1^) and maximum in the region of ~1215 cm^−1^, corresponding to peaks in the single-phonon density of states of diamond, are marked.

**Figure 4 nanomaterials-13-03124-f004:**
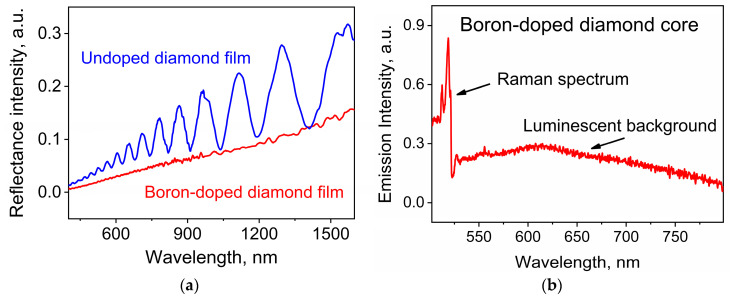
(**a**) Reflection spectra from diamond films: undoped (blue) and heavily doped with boron (red). The absence of interference fringes in the reflection spectra of the doped film is due to the significant absorption of radiation incident and reflected from the inner surface of the film due to the high boron content in the film. (**b**) Secondary emission of BND with a very weak luminescence background as compared to BND Raman peak at ~1213 cm^−1^ (~518 nm). Excitation by light with by wavelength of ~488 nm.

**Figure 5 nanomaterials-13-03124-f005:**
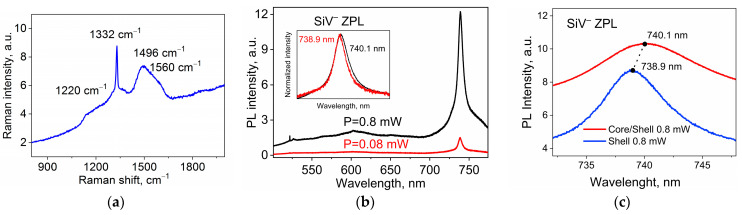
Raman and PL spectra of single core/shell ND with BND core and shell doped with SiV centers. Excitation wavelength is 488 nm. (**a**) Raman spectrum demonstrates several bands characteristic for slightly disordered lattice of diamond shell and a weak broadband of about 1220 cm^−1^ from the BND core. The Raman shifts in the band are shown. (**b**) PL spectra of the core/shell diamond nanocrystal at two strongly different excitation powers of 0.08 and 0.8 mW, resulting in the spectral shift in the ZPL. Inset illustrates the ZPL shift from 738.9 nm to 740.1 nm. (**c**) A comparison between spectral positions of the SiV ZPLs for core/shell ND with absorbing BND core (red line) and ND without boron doping (blue line) obtained at the same excitation power of 0.8 mW.

**Figure 6 nanomaterials-13-03124-f006:**
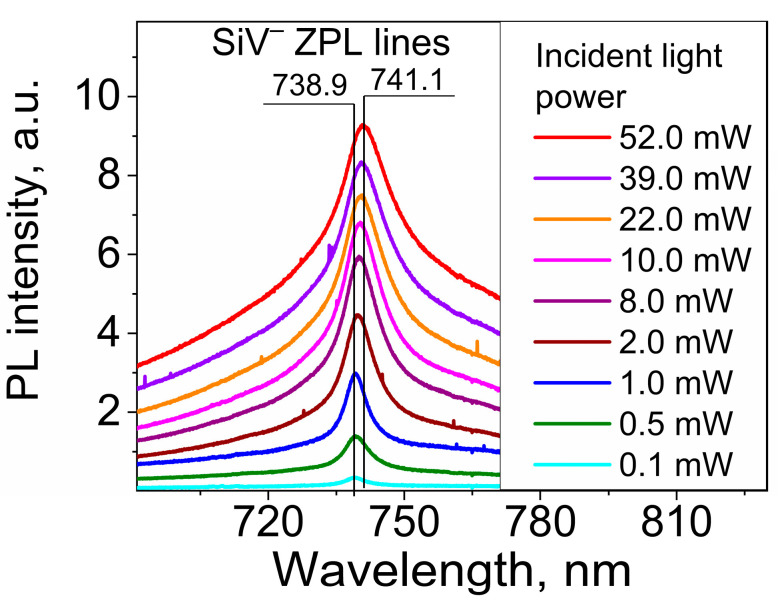
A set of PL spectra in the region of the ZPL of SiV centers excited by 532 nm radiation with different powers exerted on the sample. The vertical lines show maximum red shift in the ZPL peak with the increasing incident light power.

**Figure 7 nanomaterials-13-03124-f007:**
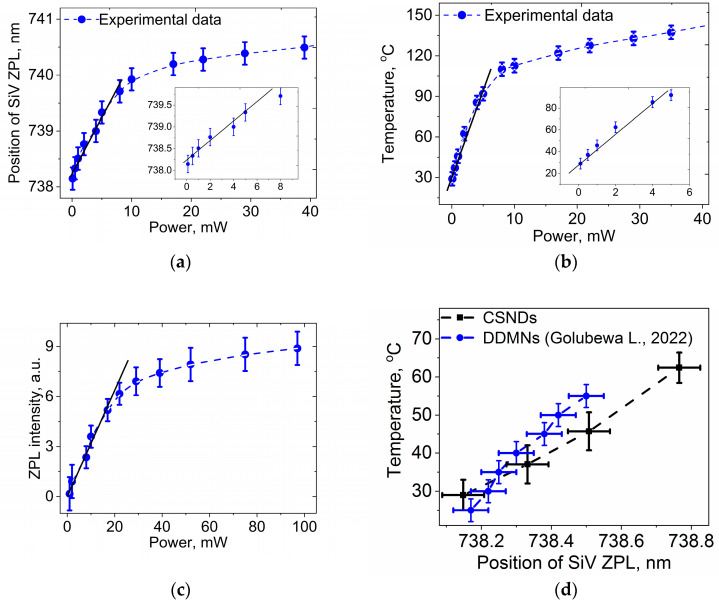
(**a**) Dependence of the spectral position of SiV ZPL in CSNDs on the laser power; the inset shows the region of linear dependence. (**b**) The dependence of the core/shell ND temperature on the laser power, calculated using data on dependence of the SiV ZPL position on temperature from [20]; the inset shows the region of linear dependence. (**c**) Dependence of SiV ZPL intensity of SiV center in CSNDs on laser radiation power, the region of linear dependence is shown. (**d**) Dependence of core/shell ND temperature on SiV ZPL position in actual power range for biological applications (black points). The obtained dependence is close to those obtained in the work of L. Golubewa et al. [17] (blue points). Lines in insets are shown for convenience.

## Data Availability

All relevant data generated and analyzed during this study, which include experimental, spectroscopic, and electron microscopic data, are included in this article.

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
