# Peer review of "Multifunctional Core/Shell Diamond Nanoparticles Combining Unique Thermal and Light Properties for Future Biological Applications"

_nanomaterials, 2023, doi:10.3390/nano13243124_

Round 1

Reviewer 1 Report

Comments and Suggestions for Authors

The manuscript by Bogdanov et al. describes fabrication of core/shell diamond nanoparticles doped with boron for application in multiple areas including hyperthermia, thermometry and fluorescence imaging. Following are the comments related to manuscript-

The tile of paper is misleading, as no direct application of the material is provided in the field of imaging or hyperthermia. The authors should show some results related to hyperthermia/photothermal therapy and/or fluorescence imaging before asserting the usefulness of their material useful. 

The experimental evidence provided for the confirmation of boron doping of diamond nanoparticles is not sufficiently convincing. The as-claimed “core/shell” nanostructure is not apparent from SEM images shown in Figure1 and 2. Authors are advised to try elemental mapping and/or TEM to prove boron doping of diamond nanoparticles.

The Raman spectra shown in Figure 3 should be compared with a control group consisting of un-doped diamond nanoparticles. This comparison would provide more robust evidence and facilitate a better understanding of the effects of boron doping on the Raman characteristics of the nanoparticles.

Why does boron doping cause quenching of fluorescence? Besides, the fluorescence peak shown in Figure 4b appears to be quite noisy. To ascertain whether the material is suitable for fluorescence imaging, the authors are advised to calculate PLQY accurately.

The presentation of figures is not optimal. For example, the font type and size of the axes in Figure 7 a,b,c,d are dissimilar. It is advisable to ensure consistent font type and size throughout the figures to improve their clarity and readability.

The captions of the figures are excessively written, providing detailed descriptions of the results. It is recommended to revise the captions to only mention the figure without going into extensive details about the results.

The introduction section should clearly emphasize the benefits of utilizing inorganic nanoparticles instead of organic nanoparticles in the field of imaging and phototherapy. This should be supported by suitable references from the pertinent literature (For e.g, Molecules 2023, 28(16), 6038, Microchimica Acta 2022, 189, 83).

Comments on the Quality of English Language

There are some grammatical and typographical errors throughout the manuscript that require correction.

Author Response

Thanks for useful comments about our manuscript. All response in attached file. Best regards.

Reviewer 2 Report

Comments and Suggestions for Authors

The paper is interesting and could be accepted after revision

Comments on the Quality of English Language

The paper is interesting and could be accepted after revision

Author Response

Thanks for useful comments about our manuscript.

In the most cases in this work correctly named silicon, only in one place where we talk about SiO2 and we use the appropriate name silica there. We also add some correction which were marked in a file.

Best regards.

Reviewer 3 Report

Comments and Suggestions for Authors

Journal: Nanomaterials

Title: Multifunctional Core/shell Diamond Nanoparticles for Local Photoinduced Hyperthermia Therapy, Thermometry, and Fluorescent Imaging

Authors have successfully synthesized the multifunctional core/shell chemical vapor deposition diamond nanoparticles for local photoinduced hyperthermia, and thermal optical sensory detection.  The spectral shifts are vivid due to the localize heating caused by the core/shell diamond nanoparticles.  By feasible modulation of the laser power, the present work extends its application towards hyperthermia therapy, and local nanothermometry. The present work is satisfactory, but the manuscript lags with some aspects and few technical discussions. Considering the above, I request authors to address the following comments, thereby suggesting the major revision for the present stage.

I suggest authors to address the following comments,

1.         Authors should consider revising the title, since the presented manuscript have not furnished any fluorescent imaging application.

2.         Authors should consider adding the scientific evidence for non-radiative recombination shifts.

3.         Since, authors have mentioned that the as-prepared core/shell materials can be utilized for therapeutics, authors can study the biocompatibility.

4.         Why single nanoparticle system is considered, why not colloidal dispersion of nanoparticles are taken into account?

5.         How these nanoparticles can be removed from the silica wafer? How is it possible to achieve the dispersion of nanoparticles in the biological system?

6.         Why specifically this dopant is chosen, what would be the effects on choosing the other possible dopants?

7.         Authors should take special emphasis on Figure number and its relevancy in the discussion part. Figure illustrations and its corresponding discussion are not aligned.

8.         Authors failed to discuss the heating rate and relaxation rate of therapeutic activity.

9.         It is highly valued to discuss the preparation cost analysis, which will showcase the importance of the presented research.

10.     To study the hyperthermia therapeutical effects, authors can use FLIR camera to know the temperature evolution over time.

11.     Graphical illustrations are inconsistent with font style and consistent axis labels, which reduces the readability.

12.     I suggest authors to refer the below articles and cite the following recent literatures to strengthen the manuscript quality,

Adv. Mater. 2023, 35, 2303267. https://doi.org/10.1002/adma.202303267

Analyst, 2023,148, 3918-3930. https://doi.org/10.1039/D3AN00800B

ACS Appl. Mater. Interfaces 2023, 15, 38, 44607–44620. https://doi.org/10.1021/acsami.3c05297 

Applied Surface Science, 516, 2020, 145661. https://doi.org/10.1016/j.apsusc.2020.145661

All the best for the revision. After receiving the convincing point-to-point responses, I will recommend it for the publication.

Comments on the Quality of English Language

I request authors to focus on minor errors like abbreviation, spelling errors and formatting errors, as well as grammatical errors.

Author Response

Thanks for the comments, all edits and answers are in the attached file.
Best regards.

Reviewer 4 Report

Comments and Suggestions for Authors

The authors had shown an interesting study entitled Multifunctional Core/shell Diamond Nanoparticles for Local 2 Photoinduced Hyperthermia Therapy, Thermometry, and Fluorescent Imaging. I think that the manuscript can be accepted for its publication in Nanomaterials, but can be improved with the next comments:

- All the chemical formulas are incorrect: the subscripts are missing.

- Correct superscripts in various parts of the text.

- Line 87: Change the phrase reference 23…. since it is the same authors who proposed that technique.

- Line 104: It should be indicated whether the treatment with hydrogen fluoride (HF) and ammonium fluoride NH4F change the morphology of nanoparticles: a TEM image should be adequate.

- Line 186: Are the authors referring to different batches?

- Line 235: I think that section 3.1.3. It should be re-written; I think there are figures that do not agree with the text.

- Line 263-277: The explanation of figure 5 is confusing; I think there is some wrong information.

- The references do not conform to the Nanomaterials format: indicate all authors, position of the year of publication, etc.

Author Response

Thanks for the comments, all edits and answers are in the attached file.
Best wishes.

Round 2

Reviewer 1 Report

Comments and Suggestions for Authors

The authors have adequately addressed most of my previous concerns. The current title of manuscript is also appropriate. I  recommend the current version of manuscript for publication.

Comments on the Quality of English Language

Minor changes are required which can be done during the proof reading process

Reviewer 3 Report

Comments and Suggestions for Authors

I agree to publish this article in the present form. Authors have addressed the queries and I wish them good luck.

Comments on the Quality of English Language

It is necessary to proofread and organize the content wisely with the right descriptive flow.